# HIERARCHICAL PROBABILISTIC MODEL FOR BLIND SOURCE SEPARATION VIA LEGENDRE TRANSFORMATION

## ABSTRACT

We present a novel *blind source separation* (BSS) method, called *information geometric blind source separation* (IGBSS). Our formulation is based on the log-linear model equipped with a hierarchically structured sample space, which has theoretical guarantees to uniquely recover a set of source signals by minimizing the KL divergence from a set of mixed signals. Source signals, received signals, and mixing matrices are realized as different layers in our hierarchical sample space. Our empirical results have demonstrated on images and time series data that our approach is superior to well established techniques and is able to separate signals with complex interactions.

## 1 INTRODUCTION

The objective of *blind source separation* (BSS) is to identify a set of source signals from a set of multivariate mixed signals[1]. BSS is widely used for applications which are considered to be the "cocktail party problem". Examples include image/signal processing (Isomura & Toyoizumi, 2016), artifact removal in medical imaging (Vigário et al., 1998), and electroencephalogram (EEG) signal separation (Congedo et al., 2008). Currently, there are a number of solutions for the BSS problem. The most widely used approaches are variations of principal component analysis (PCA) (Pearson, 1901; Murphy, 2012) and independent component analysis (ICA) (Comon, 1994; Murphy, 2012). However, they all have limitations with their approaches.

PCA and its modern variations such as sparse PCA (SPCA) (Zou et al., 2006), non-linear PCA (NLPCA) (Scholz et al., 2005), and Robust PCA (Xu et al., 2010) extract a specified number of components with the largest variance under an orthogonal constraint, which are composed of a linear combination of variables. They create a set of uncorrelated orthogonal basis vectors that represent the source signal. The basis vectors with the $N$ largest variance are called the principal components and is the output of the model. PCA has shown to be effective for many applications such as dimensionality reduction and feature extraction. However, for BSS, PCA makes the assumption that the source signals are orthogonal, which is often not the case in most practical applications.

Similarly, ICA also attempts to find the $N$ components with the largest variance, but relaxes the orthogonality constraint. All variations of ICA such as infomax (Bell & Sejnowski, 1995), FastICA (Hyvärinen & Oja, 2000) and JADE (Cardoso, 1999) separate a multivariate signal into additive subcomponents by maximizing statistical independence of each component. ICA assumes that each component is non-gaussian and the relationship between the source signal and the mixed signal is an affine transformation. In addition to these assumptions, ICA is sensitive to the initialization of the weights as the optimization is non-convex and is likely to converge to a local optimum.

Other potential methods which can perform BSS include non-negative matrix factorization (NMF) (Lee & Seung, 2001; Berne et al., 2007), dictionary learning (DL) (Olshausen & Field, 1997), and reconstruction ICA (RICA) (Le et al., 2011). NMF, DL and RICA are degenerate approaches to recover the source signal from the mixed signal. These approaches are more typically used for feature extraction. NMF factorizes a matrix into two matrices with nonnegative elements representing weights and features. The features extracted by NMF can be used to recover the source

---

[1]Mixed signals and received signals are used exchangeably throughout this article.

signal. More recently there are more advanced techniques that uses Short-time Fourier transform (STFT) to transform the signal into the frequency domain to construct a spectrogram before applying NMF (Sawada et al., 2019). However, NMF does not maximize statistical independence which is required to completely separate the mixed signal into the source signal, and it is also sensitive to initialization as the optimization is non-convex. Due to the non-convexity, additional constraints or heuristics for weight initialization is often applied to NMF to achieve better results (Ding et al., 2008; Boutsidis & Gallopoulos, 2008). DL can be thought of as a variation of the ICA approaches which requires an over-complete basis vector for the mixing matrix. DL may be advantageous because additional constraints such as a positive code or a dictionary can be applied to the model. However, since it requires an over-complete basis vector, information may be lost when reconstructing the source signal. In addition, like all the other approaches, DL is also non-convex and it is sensitive to the initialization of the weights.

All previous approaches have limitations such as loss of information or non-convex optimization and require constraints or assumptions such as orthogonality or an affine transformation which are not ideal for BSS. In the following, we introduce our approach to BSS, called IGBSS (Information Geometric BSS), using the *log-linear model* (Agresti, 2012), which can introduce relationships between possible states into its sample space (Sugiyama et al., 2017). Unlike the previous approaches, our proposed approach does not have the assumptions or limitations that they require. We provide a flexible solution by introducing a *hierarchical structure* between signals into our model, which allows us to treat interactions between signals that are more complex than an affine transformation. Unlike other existing methods, our approach does not require the inversion of the mixing matrix and is able to recover the sign of the signal. Thanks to the well-developed information geometric analysis of the log-linear model (Amari, 2001), optimization of our method is achieved via convex optimization, hence it always arrives at the globally optimal unique solution. Moreover, we theoretically show that it always minimizes the Kullback–Leibler (KL) divergence from a set of mixed signals to a set of source signals. Our experimental results demonstrate that our hierarchical model leads to better separation of signals including complex interaction such as higher-order feature interactions (Luo & Sugiyama, 2019) than existing methods.

## 2 FORMULATION

BSS is formulated as a function $f$ that separates a set of *received signals* $X$ into a set of *source signals* $Z$, i.e., $Z = f(X)$. For example, if one employs ICA based formulation, the BSS problem reduces to $\mathbf{X} = \mathbf{AZ}$, where the received signal $\mathbf{X} \in \mathbb{R}^{L \times M}$ with $L$ signals with the sample size $M$ is affine transformation of the source signal $\mathbf{Z} \in \mathbb{R}^{N \times M}$ with $N$ signals and a mixing matrix $\mathbf{A} \in \mathbb{R}^{L \times N}$. The objective is to estimate $\mathbf{Z}$ by learning $\mathbf{A}$ given $\mathbf{X}$. Our idea is to use the *log-linear model* (Agresti, 2012), which is a well-known energy-based model, to take non-affine transformation into account and formulate BSS as a convex optimization problem.

### 2.1 LOG-LINEAR MODEL ON PARTIALLY ORDERED SET

We use the log-linear model given in the form of

$$\log p(\omega) = \sum_{s \in \mathcal{S}} \mathbf{1}_{s \preceq \omega} \theta_s - \psi(\theta), \tag{1}$$

where $p(\omega) \in (0, 1)$ is probability of each state $\omega \in \Omega$ and $\mathcal{S} \subseteq \Omega$ is a parameter space such that a parameter value $\theta_s \in \mathbb{R}$ is associated with each $s \in \mathcal{S}$, and $\psi(\theta)$ is the partition function so that $\sum_{\omega \in \Omega} p(\omega) = 1$. In this formulation, we assume that the set $\Omega$ of possible states, equivalent to the sample space in the statistical sense, is a *partially ordered set* (poset); that is, it is equipped with a partial order "$\preceq$" (Gierz et al., 2003) and $\mathbf{1}_{s \preceq \omega} = 1$ if $s \preceq \omega$ and 0 otherwise. This formulation is firstly introduced by Sugiyama et al. (2016) and used to model the matrix balancing problem (Sugiyama et al., 2017), which includes Boltzmann machines as a special case (Luo & Sugiyama, 2019). If we index $\Omega$ as $\Omega = \{\omega_1, \omega_2, \ldots, \omega_{|\Omega|}\}$, we obtain the following matrix form:

$$\log \boldsymbol{p} = \mathbf{F}\boldsymbol{\theta} - \boldsymbol{\psi}(\theta),$$

where $\boldsymbol{p} \in (0, 1)^{|\Omega|}$ with $p_i = p(\omega_i)$, $\boldsymbol{\theta} \in \mathbb{R}^{|\Omega|}$ such that $\theta_i = \theta_{\omega_i}$ if $\omega_i \in \mathcal{S}$ and $\theta_i = 0$ otherwise, $\mathbf{F} = (f_{ij}) \in \{0, 1\}^{|\Omega| \times |\Omega|}$ with $f_{ij} = \mathbf{1}_{\omega_j \preceq \omega_i}$, and $\boldsymbol{\psi}(\theta) = (\psi(\theta), \ldots, \psi(\theta)) \in \mathbb{R}^{|\Omega|}$. Each vector

is treated as a column vector, and $\log$ is entry-wise operation. This matrix form is often used as a general form of the log-linear model (Coull & Agresti, 2003) and $\mathbf{F}$ is called a *model matrix*, which represents relationship between states. The assumption to the log-linear model is that $\mathbf{F}$ is needed to be non-singular, and Sugiyama et al. (2017) showed that Equation (1) with a poset $\Omega$ always provides a non-singular model matrix; that is, $\mathbf{F}$ is regular as long as each entry is given as $f_{ij} = \mathbf{1}_{\omega_j \preceq \omega_i}$. This property is powerful in mathematical modeling as we can introduce any partial order structure into $\Omega$, which we will use to introduce our hierarchical structure in tne next subsection to solve BSS.

## 2.2 LAYER CONFIGURATION FOR BLIND SOURCE SEPARATION

Our key idea is to introduce a hierarchical layered structure into the sample space $\Omega$ of the log-linear model to achieve BSS. We call this model *information geometric BSS* (IGBSS) as its optimality is supported by the tight connection between the log-linear model and information geometric property of the space of distributions (statistical manifold), which will be shown in the next subsection. We implement three layers of BSS, the mixing layer, the source layer, and the received layer, into $\Omega$ as partial orders and learn the joint representation on it using the log-linear model. The received layer and the source layer represent the input received signal and the output source signal of BSS, respectively, and the mixing layer encodes information of how to mix the source signal. In the following, we consistently assume that $L$ is the number of received signals, $M$ is the sample size, and $N$ is the number of source signals.

Let us construct three layers in the sample space $\Omega$ as $\Omega = \{\bot\} \cup \mathcal{A} \cup \mathcal{Z} \cup \mathcal{X}$ with $\mathcal{A} = \{a_{11}, \ldots, a_{LN}\}$, $\mathcal{Z} = \{z_{11}, \ldots, z_{NM}\}$, and $\mathcal{X} = \{x_{11}, \ldots, x_{LM}\}$. The element $\bot$ denotes the least element, and it acts as a partition function and $\theta_\bot = -\psi(\theta)$ always holds. We use 2D indexing of elements in each layer to make the correspondence between our formulation and ICA based formulation clear; that is, these three layers $\mathcal{A}$, $\mathcal{Z}$, and $\mathcal{X}$ are analogue to a mixing matrix $\mathbf{A} \in \mathbb{R}^{L \times N}$, a source matrix $\mathbf{Z} \in \mathbb{R}^{N \times M}$, and a received matrix $\mathbf{X} \in \mathbb{R}^{L \times M}$, respectively[2]. We will also use symbols $\omega$ and $s$ to denote elements of $\Omega$, i.e., they can be $\bot$, $a_{ln}$, $z_{nm}$, and $x_{lm}$. It is always assumed that the parameter space of the log-linear model $\mathcal{S} = \mathcal{A} \cup \mathcal{Z} \subset \Omega$, meaning that mixing and source layers are used as parameters to represent distributions in our model. Here we introduce a partial order $\preceq$ between layers. Define

$$\begin{bmatrix} a_{11} & a_{12} \\ a_{21} & a_{22} \end{bmatrix} \begin{bmatrix} z_{11} & z_{12} \\ z_{21} & z_{22} \end{bmatrix} = \begin{bmatrix} x_{11} & x_{12} \\ x_{21} & x_{22} \end{bmatrix}$$

Figure 1: An example of our sample space. Dashed lines show removed partial orders to allow for learning.

$$\begin{cases} a_{ij} \preceq z_{i'j'} & \text{if } j = i', \\ a_{ij} \npreceq z_{i'j'} & \text{otherwise,} \end{cases} \qquad \begin{cases} z_{ij} \preceq x_{i'j'} & \text{if } j = j', \\ z_{ij} \npreceq x_{i'j'} & \text{otherwise} \end{cases} \tag{2}$$

for each element in three layers $\mathcal{A}$, $\mathcal{Z}$, and $\mathcal{X}$, and we do not any ordering among elements in the same layer. Since it is a partial order, transitivity always holds, for example, $a_{11} \preceq x_{22}$ as $a_{11} \preceq z_{12}$ and $z_{12} \preceq x_{22}$. The first condition encodes the structure such that the source layer is higher than the mixing layer, and the second condition encodes that the received layer is higher than the source layer. An example of our sample space with $L = M = N = 2$ is illustrated in Figure 1.

The joint distribution for BSS is described by the log-linear model in Equation (1) over the sample space $\Omega = \{\bot\} \cup \mathcal{A} \cup \mathcal{Z} \cup \mathcal{X}$ equipped with the partial order defined in Equation (2). If we learn the joint distribution from a received signal $\mathbf{X}$, we will obtain probabilities on the source layer $p(z_{11}), \ldots, p(z_{NM})$, which represents normalized source signals. The rational of our approach is given as follows: The connections between each layer is structured so that the log-linear model performs a similar computation with the ICA based approach $\mathbf{X} = \mathbf{A}\mathbf{Z}$. Our structure ensures that each $p(x_{lm})$ is determined by $(\theta_{a_{ln}})_{n \in [N]}$ and $(\theta_{z_{mn}})_{n \in [N]}$ with $[N] = \{1, \ldots, N\}$, as we always have $a_{ln} \preceq x_{lm}$ and $z_{nm} \preceq x_{lm}$. Moreover, this formulation allows us to model more complex interaction than affine transformation, such as higher-order interactions, between signals if

---

[2]We abuse an entry $x_{lm}$ of $\mathbf{X}$ and its corresponding state in $\mathcal{X}$ to avoid complicated notations.

we additionally include partial order structure into $\mathcal{Z}$ and/or $\mathcal{A}$, which cannot be treated by a simple matrix multiplication.

## 2.3 OPTIMIZATION

We train the log-linear model by minimizing the KL divergence from an empirical distribution $\hat{p}$, which is identical to the normalized received signal $\mathbf{X} \in \mathbb{R}^{L \times M}$, to the model joint distribution $p$ given by Equation (1) or, equivalently, maximizing the likelihood. More precisely, we normalize a given $\mathbf{X}$ by dividing each entry by the sum of all entries; that is, an empirical distribution $\hat{p}$ is obtained as $\hat{p}(x_{lm}) = x_{lm}/\sum_{l,m} x_{lm}$. If $\mathbf{X}$ contains negative values, an exponential kernel $\exp{(x_{lm})}/\sum_{l,m} \exp{(x_{lm})}$ or min-max normalization $(x_{lm}+\epsilon-\min(\mathbf{X}))/(\max(\mathbf{X})+\epsilon-\min(\mathbf{X}))$ can be used, where $\epsilon$ is some arbitrary small value to avoid zero probability. We also assume that $\hat{p}(a_{ln}) = 0$ and $\hat{p}(z_{nm}) = 0$ for all $a_{ln} \in \mathcal{A}$ and $z_{nm} \in \mathcal{Z}$. The objective function is given as

$$\underset{p \in \mathfrak{P}_\theta}{\arg\min}\, \mathrm{D}_{\mathrm{KL}}\,(\hat{p}\|p) = \underset{p \in \mathfrak{P}_\theta}{\arg\min} \sum_{\omega \in \Omega} \hat{p}(\omega) \log \frac{\hat{p}(\omega)}{p(\omega)}, \tag{3}$$

where $\mathfrak{P}_\theta$ is the set of distributions that can be represented by Equation (1) with our structured sample space $\Omega = \{\bot\} \cup \mathcal{A} \cup \mathcal{Z} \cup \mathcal{X}$ and $\mathcal{S} = \mathcal{A} \cup \mathcal{Z}$.

The remarkable property of our model is that *this optimization problem is convex* and it is guaranteed that gradient-based methods can always arrive at the globally optimal unique solution. To show this, we analyze the geometric structure of the statistical manifold, the set of probability distributions, generated by the log-linear model. Let $\Omega^+ = \Omega \setminus \{\bot\}$. First we introduce another parameterization $(\eta_\omega)_{\omega \in \Omega^+}$ of the log-linear model, which is defined as

$$\eta_\omega = \sum_{s \in \Omega} \mathbf{1}_{\omega \preceq s}\, p(s). \tag{4}$$

Note that $\eta_\bot = 1$ always holds and we do not include it into parameters. In addition, for theoretical consistency we change the parameter space used in Equation (1) from $\mathcal{S}$ to $\Omega^+$ and assume that $\theta_\omega = 0$ if $\omega \notin \mathcal{S}$. Again we do not include $\theta_\bot$ as a parameter as it is the partition function. Two parameters $(\theta_\omega)_{\omega \in \Omega^+}$ and $(\eta_\omega)_{\omega \in \Omega^+}$ have clear statistical interpretation as it is widely known that any log-linear model belongs to the exponential family, where $\theta$ and $\eta$ correspond to natural and expectation parameters, respectively. $\theta$ and $\eta$ are connected via a Legendre transformation which means that they are both differentiable and have a one-to-one correspondence. To simplify the notation, we denote by $\hat{\theta}$ and $\hat{\eta}$ the corresponding $\theta$ and $\eta$ of the empirical distribution $\hat{p}$. Let $\mathfrak{P} = \{p \mid 0 < p(\omega) < 1$ for all $\omega \in \Omega\}$ be the set of all probability distributions. This set forms a statistical manifold with dually flat structure, which is the canonical geometric structure in information geometry (Amari, 2016), with its dual coordinate system $((\theta_\omega)_{\omega \in \Omega^+}, (\eta_\omega)_{\omega \in \Omega^+})$; that is, both of $(\theta_\omega)_{\omega \in \Omega^+}$ and $(\eta_\omega)_{\omega \in \Omega^+}$ work as coordinate systems and determine a distribution in $\mathfrak{P}$. The Riemannian metric with respect to $\theta$ is given as

$$g_{ss'} = \frac{\partial \eta_s}{\partial \theta_{s'}} = \mathbb{E}\left[\frac{\partial \log p(\omega)}{\partial \theta_s} \frac{\partial \log p(\omega)}{\partial \theta_{s'}}\right] = \sum_{\omega \in \Omega} \mathbf{1}_{s \preceq \omega} \mathbf{1}_{s' \preceq \omega}\, p(\omega) - \eta_s \eta_{s'}, \tag{5}$$

which coincides with the Fisher information (Sugiyama et al., 2017, Theorem 3) and we will use it for natural gradient.

Now we consider two submanifolds $\mathfrak{P}_\theta, \mathfrak{P}_\eta \subseteq \mathfrak{P}$, which we define as

$$\begin{aligned} \mathfrak{P}_\theta &= \{\, p \in \mathfrak{P} \mid \theta_\omega = 0, \forall \omega \in \mathcal{E} \,\}, & \mathcal{E} &= \Omega^+ \setminus \mathcal{S}, \\ \mathfrak{P}_\eta &= \{\, p \in \mathfrak{P} \mid \eta_\omega = \hat{\eta}_\omega, \forall \omega \in \mathcal{M} \,\}, & \mathcal{M} &= \mathcal{S}. \end{aligned}$$

Note that this $\mathfrak{P}_\theta$ coincides with that in Equation (3). The submanifold $\mathfrak{P}_\theta$ is called an *e-flat* submanifold and $\mathfrak{P}_\eta$ an *m-flat* submanifold in information geometry. The highlight of considering these two types of submanifolds is that, if $\mathcal{E} \cap \mathcal{M} = \emptyset$ and $\mathcal{E} \cup \mathcal{M} = \Omega^+$, it is theoretically guaranteed that the intersection $\mathfrak{P}_\theta \cap \mathfrak{P}_\eta$ is always a singleton and it is the optimizer of Equation (3) (Amari, 2009, Theorem 3), that is, it is the globally optimal solution of our model.

Optimization is achieved by *e-projection*, which seeks $\mathfrak{P}_\theta \cap \mathfrak{P}_\eta$ in the *e*-flat submanifold $\mathfrak{P}_\theta$. The *e*-projection is always *convex optimization* as $\mathfrak{P}_\theta$ is convex with respect to $\theta$; this is because $\theta$ is

---

**Algorithm 1** Information Geometric BSS

---

1: **Function** IGBSS($\mathbf{X}$, $\mathcal{S}$):
2: Compute $\hat{p}$ from $\mathbf{X}$
3: Compute $\hat{\boldsymbol{\eta}} = (\hat{\eta}_s)_{s \in \mathcal{S}}$ from $\hat{p}$
4: Initialize $(\theta_s)_{s \in \mathcal{S}}$ (randomly or $\theta_s = 0$)
5: **repeat**
6:     Compute $p$ using the current parameter $(\theta_s)_{s \in \mathcal{S}}$
7:     Compute $(\eta_s)_{s \in \mathcal{S}}$ from $p$
8:     $(\Delta \eta_\omega)_{\omega \in \mathcal{Z}} \leftarrow (\eta_\omega)_{\omega \in \mathcal{Z}} - (\hat{\eta}_\omega)_{\omega \in \mathcal{Z}}$
9:     $(\Delta \eta_\omega)_{\omega \in \mathcal{A}} \leftarrow (\eta_\omega)_{\omega \in \mathcal{A}} - (\hat{\eta}_\omega)_{\omega \in \mathcal{A}}$
10:    Compute the Fisher information matrix for source layer $\mathbf{G}_Z$ and the mixing layer $\mathbf{G}_A$
11:    $(\theta_\omega)_{\omega \in \mathcal{Z}} \leftarrow (\theta_\omega)_{\omega \in \mathcal{Z}} - \mathbf{G}_Z^{-1}(\Delta \eta_\omega)_{\omega \in \mathcal{Z}}$
12:    $(\theta_\omega)_{\omega \in \mathcal{A}} \leftarrow (\theta_\omega)_{\omega \in \mathcal{A}} - \mathbf{G}_A^{-1}(\Delta \eta_\omega)_{\omega \in \mathcal{A}}$
13: **until** convergence of $(\theta_s)_{s \in \mathcal{S}}$
14: **End Function**

---

a coordinate system of $\mathfrak{P}_\theta$ that is linearly constrained on $\theta$. We can therefore use the standard gradient descent strategy to optimize the log-linear model. The derivative of the KL divergence with respect to $\theta_s$ is known to be the difference between expectation parameters $\eta$ (Sugiyama et al., 2017, Theorem 2): $(\partial / \partial \theta_s) D_{\mathrm{KL}}(\hat{p} \, \| \, p) = \eta_s - \hat{\eta}_s$, and the KL divergence $D_{\mathrm{KL}}(\hat{p} \| p)$ is minimized if and only if $\eta_s = \hat{\eta}_s$ for all $s \in \mathcal{S}$.

From our definition of $\Omega$ in Equation (2), we have $\eta_{z_{kl}} = \eta_{z_{k'l}}$ for all $z_{kl}, z_{k'l} \in \mathcal{Z}$. Therefore all elements in the source layer will learn the same value. This problem can be avoided by removing some of partial orders between source and received layers. We propose to systematically remove the partial order $z_{ij} \preceq x_{i'j'}$ if $i = i'$ to ensure $\eta_{z_{kl}} \neq \eta_{z_{k'l}}$ (see Figure 1), while other strategies are possible as long as $\eta_{z_{kl}} \neq \eta_{z_{k'l}}$ is satisfied, for example, random deletion of such orders.

Using the above results, gradient descent can be directly applied to achieve Equation (3). However, this may need a large number of iterations to reach convergence. To reduce the number of iterations, we propose to use *natural gradient* (Amari, 1998), which is a second-order optimization approach and will also always find the global optimum. Let us re-index $\mathcal{S} = \mathcal{A} \cup \mathcal{Z}$ as $\mathcal{S} = \{s_1, s_2, \dots, s_{|\mathcal{S}|}\}$ and assume that $\boldsymbol{\theta} = [\theta_{s_1}, \dots, \theta_{s_{|\mathcal{S}|}}]^{\mathrm{T}}$ and $\boldsymbol{\eta} = [\eta_{s_1}, \dots, \eta_{s_{|\mathcal{S}|}}]^{\mathrm{T}}$. In each step of natural gradient, the current $\boldsymbol{\theta}$ is updated to $\boldsymbol{\theta}_{\mathrm{next}}$ by the following formula:

$$\boldsymbol{\theta}_{\mathrm{next}} = \boldsymbol{\theta} - \mathbf{G}^{-1}(\boldsymbol{\eta} - \hat{\boldsymbol{\eta}})$$

where $\mathbf{G} = (g_{ij}) \in \mathbb{R}^{|\mathcal{S}| \times |\mathcal{S}|}$ is the Fisher information matrix such that each $g_{ij}$ is given as $g_{s_i s_j}$ in Equation (5).

Although the natural gradient requires much less iterations compared to the gradient descent, matrix inversion $\mathbf{G}^{-1}$ is computationally expensive as it has the complexity of $\mathcal{O}(|\mathcal{S}|^3)$. In addition, FIM values are often too small and optimization becomes numerically unstable. To solve these problems, we separate the update steps in the source layer and the mixing layer:

$$(\theta_{\omega, \mathrm{next}})_{\omega \in \mathcal{Z}} = (\theta_\omega)_{\omega \in \mathcal{Z}} - \mathbf{G}_Z^{-1}(\Delta \eta_\omega)_{\omega \in \mathcal{Z}}, \tag{6}$$

$$(\theta_{\omega, \mathrm{next}})_{\omega \in \mathcal{A}} = (\theta_\omega)_{\omega \in \mathcal{A}} - \mathbf{G}_A^{-1}(\Delta \eta_\omega)_{\omega \in \mathcal{A}}, \tag{7}$$

where $\mathbf{G}_Z$ and $\mathbf{G}_A$ are the Fisher information matrices for source and mixing layers, respectively. Note that this also leads to the same global optimum. They are constructed by assuming all the other parameters are fixed. This approach reduces the time complexity to $\mathcal{O}(|\mathcal{Z}|^3 + |\mathcal{A}|^3)$. The full algorithm using natural gradient is given in Algorithm 1. Computation of $p$ from $\theta$ and $\eta$ from $p$ can be achieved using Equations (1) and (4). We also give more explicit description of $p$ and $\eta$ for each layer in Appendix. The time complexity to compute $p$ in Algorithm 1 Line 6 is $\mathcal{O}(|\Omega||S|)$. The complexity to compute $\Delta \boldsymbol{\eta}$ in Algorithm 1 Line 8 and Line 9 is $\mathcal{O}(|\mathcal{Z}|) + \mathcal{O}(|\mathcal{A}|) = \mathcal{O}(|\mathcal{S}|)$. Therefore the total complexity of each iteration is $\mathcal{O}(|\mathcal{Z}|^3 + |\mathcal{A}|^3 + |\Omega||\mathcal{S}|)$.

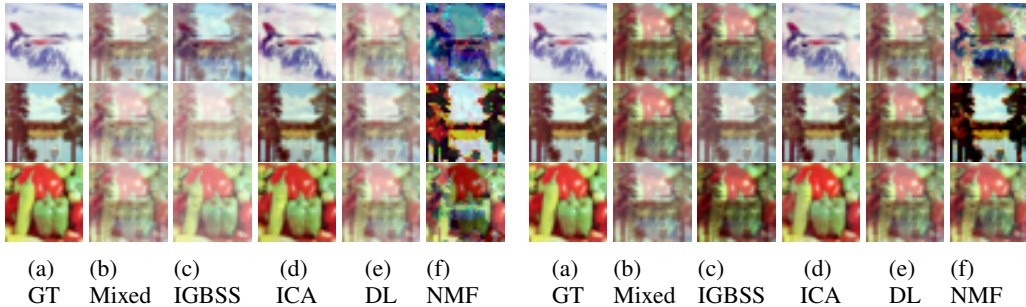

|         | (a)  | (b)   | (c)   | (d) | (e) | (f) |       | (a)  | (b)   | (c)   | (d) | (e) | (f) |
| :-----: | :--: | :---: | :---: | :-: | :-: | :-: | :---: | :--: | :---: | :---: | :-: | :-: | :-: |
|         | GT   | Mixed | IGBSS | ICA | DL  | NMF |       | GT   | Mixed | IGBSS | ICA | DL  | NMF |

Figure 2: First-order interaction experiment.  Figure 3: Third-order interaction experiment.

Table 1: Signal-to-Noise Ratio of reconstructed signal. (∗) Results for Figure 2. (†) Results for Figure 3. Scores are means ± standard deviation after 40 runs. We have applied different weight initialization after each run.

| Exp. | Order | Root Mean Squared Error (RMSE) | | | | Signal-to-noise ratio (SNR) (units in dB) | | | |
| :--: | :---: | :---: | :---: | :---: | :---: | :---: | :---: | :---: | :---: |
|      |       | IGBSS | FastICA | DL | NMF | IGBSS | FastICA | DL | NMF |
|   1  | First∗  | **0.252 ± 0.000** | 0.300 ± 0.089 | 0.394 ± 0.041 | 0.622 ± 0.000 | **12.588 ± 0.000** | 11.688 ± 4.829 | 6.810 ± 0.008 | 1.704 ± 0.000 |
|      | Second  | **0.260 ± 0.000** | 0.285 ± 0.096 | 0.441 ± 0.080 | 0.662 ± 0.000 | 10.729 ± 0.000 | **12.353 ± 4.255** | 0.526 ± 0.448 | -3.426 ± 0.000 |
|      | Third†  | **0.252 ± 0.000** | 0.260 ± 0.111 | 0.362 ± 0.030 | 0.612 ± 0.000 | 12.588 ± 0.000 | **12.922 ± 5.590** | 1.471 ± 0.358 | 0.039 ± 0.000 |
|   2  | First   | **0.133 ± 0.000** | 0.284 ± 0.064 | 0.474 ± 0.067 | 0.591 ± 0.000 | **14.215 ± 0.000** | 11.218 ± 1.964 | 2.098 ± 2.140 | -0.940 ± 0.000 |
|      | Second  | **0.256 ± 0.000** | 0.263 ± 0.066 | 0.576 ± 0.008 | 0.684 ± 0.000 | **10.612 ± 0.000** | 11.986 ± 2.157 | -1.589 ± 0.269 | -3.675 ± 0.000 |
|      | Third   | **0.282 ± 0.000** | 0.239 ± 0.056 | 0.593 ± 0.007 | 0.665 ± 0.000 | 9.346 ± 0.000 | **11.475 ± 2.145** | -2.274 ± 0.227 | -4.073 ± 0.000 |
|   3  | First   | **0.155 ± 0.000** | 0.699 ± 0.047 | 0.478 ± 0.121 | 0.628 ± 0.000 | **11.285 ± 0.000** | 10.785 ± 2.176 | 1.448 ± 4.249 | 0.628 ± 0.000 |
|      | Second  | **0.200 ± 0.000** | 0.280 ± 0.049 | 0.515 ± 0.007 | 0.709 ± 0.000 | **10.862 ± 0.000** | 10.171 ± 2.353 | 0.529 ± 0.228 | -5.579 ± 0.000 |
|      | Third   | **0.203 ± 0.000** | 0.239 ± 0.056 | 0.536 ± 0.006 | 0.682 ± 0.000 | **11.075 ± 0.000** | 11.041 ± 2.708 | -0.244 ± 0.185 | -4.961 ± 0.000 |

## 3 EXPERIMENTS

We empirically examine the effectiveness of IGBSS to perform BSS using real-world image and synthetic time-series datasets for an affine transformation and higher-order interactions between signals. All experiments were run on CentOS Linux 7 with Intel Xeon CPU E5-2623 v4 and Nvidia QuadroGP100 [3].

### 3.1 BLIND SOURCE SEPARATION FOR AFFINE TRANSFORMATIONS ON IMAGES

In our experiments, we use three benchmark images widely used in computer vision from the University of Southern California's Signal and Image Processing Institute (USC-SIPI)[4], which include "airplane (F-16)", "lake" and "peppers". Each image is standardized to have 32x32 pixels with red, green and blue color channels with integer values between 0 and 255 to represent the intensity of each pixel. These images shown in Figure 2a are the source signal $\mathbf{Z}$ which are unknown to the model. They are only used as ground truth to evaluate the model's output. The equation $\mathbf{X} = \mathbf{AZ}$ is used to generate the received signal $\mathbf{X}$ by randomly generating values for a mixing matrix $\mathbf{A}$ using the uniform distribution which generates real numbers between 1 and 6. The images are then rescaled to integer values within the range between 0 and 255. The received signal $\mathbf{X}$, which is the input to the model, is the three images shown in Figure 2b. The three images for the mixed signal may look visually similar, however, they are actually superposition of the source signal with different intensity. The objective of our model is to reconstruct the source signal $\mathbf{Z}$ without knowing the mixing matrix $\mathbf{A}$.

We compare our approach to FastICA (Hyvärinen & Oja, 2000) with the $\log \cosh$ function as the signal prior, dictionary learning (DL) (Olshausen & Field, 1997) with constraint for positive dictionary and positive code, and NMF with the coordinate descent solver and non-negative double singular value decomposition (NNDSVD) initialization (Boutsidis & Gallopoulos, 2008) with zero values replaced with the mean of the input.

---

[3]The code is available in the supplementary material and will be publicly available online after the peer review process

[4]http://sipi.usc.edu/database/

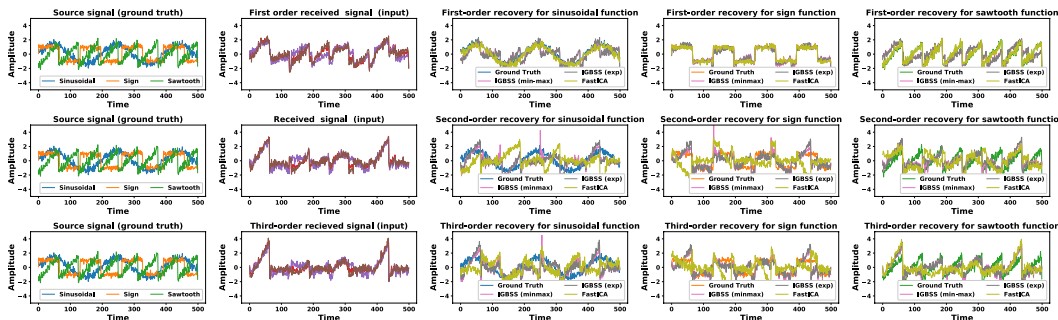

Figure 4: Time series signal experiment.

Since BSS is an unsupervised learning problem, the order of the signal is not recovered. We identify the corresponding signal by taking all permutations of the output and calculate the minimum euclidean distance with the ground truth. The permutation which returns the minimum error is considered as the correct order of the image. The scale of the output is also not recovered, thereby we have used min-max normalization to the output of each model.

Separation results for images are shown in Figure 2. Our proposed approach IGBSS is able to recover majority of the "shape" of the source signal, while the intensity of each image appears to larger than the ground truth for all images. Small residuals of each image can be seen on the other images. For instance, in the airplane (F-16) image, there residuals from the lake image can be clearly seen. Compared to the reconstruction of IGBSS with FastICA, DL and NMF, IGBSS performs significantly better as all the other approaches are unable to clearly separate the mixed signal. FastICA was unable to provide a reasonable reconstruction with 3 mixed signal. To overcome this limitation of FastICA, we randomly generated another column of the mixing matrix and append it to the current mixing matrix to create 4 mixed signals as an input to FastICA to recover a more reasonable signal.

The root mean square error (RMSE) of the Euclidean distance and the signal-to-noise ratio (SNR) between the reconstruction and the ground truth is calculated to quantify results of each method. The SNR is computed by $\mathrm{SNR}_{dB} = 20 \log_{10}(z_{\mathrm{norm}}/|(z - z_{\mathrm{norm}})|)$. The full results are shown in Table 1 (top row for each experiment). In the table, we present three experiments with different RGB images from USC-SIPI dataset, for each experiment we generate a new mixing matrix, where the second and the third experiments uses images of "mandrill", "splash", "jelly beans" and "mandrill", "lake", "peppers", respectively. Ground truth and resulting images for second and third experiments are presented in Supplement. Our results clearly show that IGBSS is superior to other methods, that is, IGBSS has consistently produced the lowest RMSE error for every experiment. When looking at the SNR ratio, our model has produce the highest SNR for majority of the cases and is always able to recover the same result after each run as it is formulated as convex optimization.

## 3.2 BLIND SOURCE SEPARATION WITH HIGHER-ORDER FEATURE INTERACTIONS

We demonstrate the ability of BSS for our model to include *higher-order feature interactions* in BSS. We use the same benchmark images in the standard BSS as the source signal $\mathbf{Z}$ for our experiment. We generate the higher-order feature interactions of the received signal by using the multiplicative product of the source signal. If we take into account up to $k$th order interaction ($k \leq N$),

$$x_{lm} = \sum_n a_{ln} z_{nm} + \sum_{n_1} \sum_{n_2 > n_1} a_{ln_1 n_2} z_{n_1 m} z_{n_2 m} + \sum_{n_1} \sum_{n_2 > n_1} \sum_{n_3 > n_2} a_{ln_1 n_2 n_3} z_{n_1 m} z_{n_2 m} z_{n_3 m}$$
$$+ \cdots + \sum_{n_1} \cdots \sum_{n_k > n_{k-1}} a_{ln_1 \ldots n_k} z_{n_1 m} \ldots z_{n_k m}.$$

All the other known approaches take into account only first order interactions (that is, affine transformation) between features. Differently, our model can directly incorporate the higher-order features as we do not have any assumption of the affine transformation. When we consider up to $k$th order interactions, we additionally include elements corresponding to new mixing parameters into the mixing layer. For example, if $k = 2$, nodes for $a_{ln_1 n_2}$ are added and $a_{ln_1 n_2} \preceq z_{nm}$ if $n_1 = n$ or

Table 2: Quantitative results for time-series separation experiment (mean $\pm$ standard deviation with 40 runs).

(a) Root Mean Squared Error (RMSE)

| Order | IGBSS (min-max) | IGBSS (exp) | FastICA |
|---|---|---|---|
| First | $0.702 \pm 0.000$ | $0.703 \pm 0.000$ | $\mathbf{0.414 \pm 0.286}$ |
| Second | $0.921 \pm 0.000$ | $\mathbf{0.921 \pm 0.000}$ | $1.700 \pm 0.167$ |
| Third | $0.967 \pm 0.000$ | $\mathbf{0.961 \pm 0.000}$ | $1.388 \pm 0.178$ |

(b) Signal-to-noise (SNR) (units in dB)

| Order | IGBSS (min-max) | IGBSS (exp) | FastICA |
|---|---|---|---|
| First | $3.596 \pm 0.000$ | $3.600 \pm 0.000$ | $\mathbf{15.391 \pm 3.813}$ |
| Second | $\mathbf{0.291 \pm 0.000}$ | $0.042 \pm 0.000$ | $-5.803 \pm 1.124$ |
| Third | $\mathbf{0.340 \pm 0.000}$ | $0.128 \pm 0.000$ | $-3.427 \pm 1.249$ |

$n_2 = n$. Figure 3 shows experimental results for the third-order feature experiment. Our approach IGBSS shows superior reconstruction of the source signal to other approaches. All the other approaches except for NMF is able to achieve reasonable reconstruction. NMF is able to recover the "shape" of the image, however, unlike IBSS, NMF is a degenerate approach, so it is unable to recover all color channels in the correct proportion, creating discoloring for the image which is clearly shown in the SNR values. Since the proportion of the intensity of the pixel is not recovered. In terms of both of the RMSE and the SNR shown in Table 1, IGBSS again shows the best results for both second- and third-order interactions of signals across three experiments.

## 3.3 TIME SERIES DATA ANALYSIS

We demonstrate the effectiveness of our model on time series data. In our experiments, we create three signals with 500 observations each using the sinusoidal function, sign function, and the sawtooth function. The synthetic data simulates typical signals from a wide range of applications including audio, medical and sensors. We randomly generate a mixing matrix by drawing from a uniform distribution with values between 0.5 and 2. In our experiment, we provide comparison of using both min-max normalization and exponential kernel as a pre-processing step and compare our approach with FastICA.

Experimental results are illustrated in Figure 4. These results show that IGBSS is superior to all the ICA approaches because it is able to recover both the shape of the signal and the sign of the signal, while all the other ICA approaches are only able to recover the shape of the signal and are unable to recover the sign of the signal. This means that ICA could recover a flipped signal. We have paired the recovered signal of ICA with the ground truth by finding the signal and sign with the lowest RMSE error. In any practical application, this is not possible for ICA because the latent signal is unknown. Through visual inspection, IGBSS is able to recover all visual signals with high accuracy, while FastICA is only able to recover the first-order interaction and it is unable to produce a reasonable recovery for second- and third-order interactions. In addition to our visual comparison, we have also performed a quantitative analysis on the experimental results using RMSE error with the ground truth. Results are shown in Table 2. FastICA has shown to have better performance for First-Order interactions. However, for second- and third-order SNR results for FastICA is unable to recover a reasonable signal because the noise is more dominant. IGBSS has shown superior performance and is able to recover the signal for second- and third-order interactions with better scores for both RMSE and SNR.

## 4 CONCLUSION

We have proposed a novel blind source separation (BSS) method, called *Information Geometric Blind Source Separation* (IGBSS). We have formulated our approach using the log-linear model, which enables us to introduce a hierarchical structure into its sample space to achieve BSS. We have theoretically shown that IGBSS has desirable properties for BSS such as unique recover of source signals as it solves the convex optimization problem by minimizing the KL divergence from mixed signals to source signals. We have experimentally shown that IGBSS recovers images and signals closer to the ground truth than independent component analysis (ICA), dictionary learning (DL) and non-negative matrix factorization (NMF). Thanks to the flexibility of the hierarchical structure, IGBSS is able to separate signals with complex interactions such as higher-order interactions. Our model is superior to the other approaches because it is non-degenerate and is able to recover the sign of the signal. Since our approach is flexible and requires less assumptions than alternative approaches, it can be applied to various real world applications such as medical imaging, signal processing, and image processing.

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

## A   APPENDIX

### A.1   PARAMETER COMPUTATION FOR EACH LAYER

In the following, we give $p$, $\eta$, and the gradient for each layer, which are used in gradient descent.

**Received Layer (Input Layer)**: Probability $p(x)$ on the received layer $x \in \mathcal{X}$ is obtained as

$$\log p(x) = \sum_{z \in \mathcal{Z}} \mathbf{1}_{z \preceq x} \theta_z + \sum_{a \in \mathcal{A}} \mathbf{1}_{a \preceq x} \theta_a + \theta_\perp, \tag{8}$$

$$\eta_x = \sum_{x' \in \mathcal{X}} \mathbf{1}_{x \preceq x'} p(x') = p(x). \tag{9}$$

We do not need to compute gradient for the received layer as there is no parameter on this layer and $\theta_x = 0$ for all $x \in \mathcal{X}$.

**Source Layer (Output Layer)**: Probability $p(z)$ on the source layer for each $z \in \mathcal{Z}$ is given as

$$\log p(z) = \sum_{z' \in \mathcal{Z},} \mathbf{1}_{z' \preceq z} \theta_{z'} + \sum_{a \in \mathcal{A}} \mathbf{1}_{a \preceq z} \theta_a + \theta_\perp = \theta_z + \sum_{a \in \mathcal{A}} \mathbf{1}_{a \preceq z} \theta_a + \theta_\perp, \tag{10}$$

$$\eta_z = \sum_{x \in \mathcal{X}} \mathbf{1}_{z \preceq x} p(x) + \sum_{z' \in \mathcal{Z}} \mathbf{1}_{z \preceq z'} p(z') = \sum_{x \in \mathcal{X}} \mathbf{1}_{z \preceq x} p(x) + p(z). \tag{11}$$

Thus the gradient for the source layer is given as

$$\frac{\partial}{\partial \theta_z} D_{KL}(\hat{p} \| p) = \eta_z - \hat{\eta}_z = \sum_{x \in \mathcal{X}} \mathbf{1}_{z \preceq x} \left( p(x) - \hat{p}(x) \right) + p(z). \tag{12}$$

**Mixing Layer**: Probability $p(a)$ on this layer for each $a \in \mathcal{A}$ is given as

$$\log p(a) = \sum_{a' \in \mathcal{A}} \mathbf{1}_{a' \preceq a} \theta_{a'} + \theta_\perp = \theta_a + \theta_\perp, \tag{13}$$

$$\eta_a = \sum_{x \in \mathcal{X}} \mathbf{1}_{a \preceq x} p(x) + \sum_{z \in \mathcal{Z}} \mathbf{1}_{a \preceq z} p(z) + \sum_{a' \in \mathcal{A}} \mathbf{1}_{a \preceq a'} p(a') \tag{14}$$

$$= \sum_{x \in \mathcal{X}} \mathbf{1}_{a \preceq x} p(x) + \sum_{z \in \mathcal{Z}} \mathbf{1}_{a \preceq z} p(z) + p(a). \tag{15}$$

The gradient of the mixing layer is given as

$$\frac{\partial}{\partial \theta_a} D_{KL}(\hat{p}\|p) = \eta_a - \hat{\eta}_a = \sum_{x \in \mathcal{X}} \mathbf{1}_{a \preceq x}\left(p(x) - \hat{p}(x)\right) + \sum_{z \in \mathcal{Z}} \mathbf{1}_{a \preceq z} p(z) + p(a). \qquad (16)$$

Parameter values $\theta_a$ in the mixing layer represent the degree of mixing between source signals. Hence they can be used to perform feature selection and extraction. For example, if $\theta_a = 0$ in the extreme case, the corresponding node $a$ does not have any contribution to the source mixing.

## A.2 Feature Extraction for a 2D Point Cloud Experiment

We demonstrate the effectiveness for IGBSS to identify independent components on a 2-dimensional point cloud to be used for feature extraction or dimensionality reduction. In our experiment, we generate a 2-dimensional point cloud using two standard Student's $t$-distribution with 1.3 degree of freedom and have scaled the first dimension by $1/5$ and the second dimension by $1/10$ to the point cloud, illustrated in Figure 5a. Then we have randomly generated a mixing matrix for our experiment to generate a mixed signal shown in Figure 5b. We run the experiment on our model IGBSS using min-max normalization as a pre-processing step and compare it to PCA and ICA. We apply the reverse transformation of the min-max normalization on the recovered signal and have plotted the results in Figure 5. From the experimental results, we can see that PCA is able to recover the same scale of the point cloud. However, the sign of the signal is not recovered as we have recovered reversed sign of the signal. PCA also recovers signals which are orthogonal to the largest variance. Therefore the axes of the point cloud recovered by PCA does not align with the source signal in Figure 5a, that is, the axes do not run parallel to the x- and y- axes but instead is still in the same orientation as the mixed signal. This is not what we want as the signal is still mixed, and we would like to recover the signal in the same orientation as the source signal in blind source separation. ICA aims to recover statistically independent signals that are generally considered as the axes with the largest variances and not necessarily orthogonal to each other. However, the limitations of ICA is that it is unable to recover the sign and the scale of the signal. Therefore the scale of the recovered signal does not match with the source signal. In our experiment, we have plotted the results with unit variance as the recovered signal is generally unnormalized in ICA. Since our experiment is synthetically generated, we are able to quantitatively measure the the error in each approach by normalizing both the recovered signal and the source signal by its standard deviation then computing the root mean squared error (RMSE) and the signal-to-noise ratio (SNR). The results of this is shown in Table 3. Our proposed approach IGBSS has clear advantages, where it is able to recover the same orientation as the source signal as well as preserve the signal.

Table 3: Signal-to-Noise Ratio (SNR) and Root Mean Square Error (RMSE) between the recovered signal and the latent source signal for the 2-dimensional point cloud experiment.

| Model | PCA | ICA | IGBSS |
|---|---|---|---|
| RMSE | 2.011 | 1.445 | **1.421** |
| SNR | 25.997 | 27.431 | **27.503** |

## A.3 Runtime Analysis

In our experiment, we used a learning rate of 1.0 for gradient descent. Although the time complexity for each iteration of natural gradient is $\mathcal{O}(|\mathcal{Z}|^3 + |\mathcal{A}|^3 + |\Omega||S|)$, which is larger than $\mathcal{O}(|\Omega||S|^2)$ for gradient descent, natural gradient is able to reach convergence faster because it is quadratic convergence and requires significantly less iterations compared to gradient descent, which linearly converges. Increasing the size of the input will increase the size of $|\Omega|$ only, while the number of parameters $|\mathcal{Z}|$, $|\mathbf{A}|$ remain this same. Since the complexity of natural gradient is linear with respect to the size $|\Omega|$ of the input, increasing the size of the input is unlikely to increase the runtime significantly. Our experimental analysis in Figure 6 supports this analysis: our model scales linearly for both natural gradient and gradient descent when increasing the order of interactions in our model. This is because for practical application it is unlikely that $|\mathcal{A}| > |\mathcal{Z}|$. The different between the runtime for natural gradient and gradient descent becomes larger as the order of interactions increased.

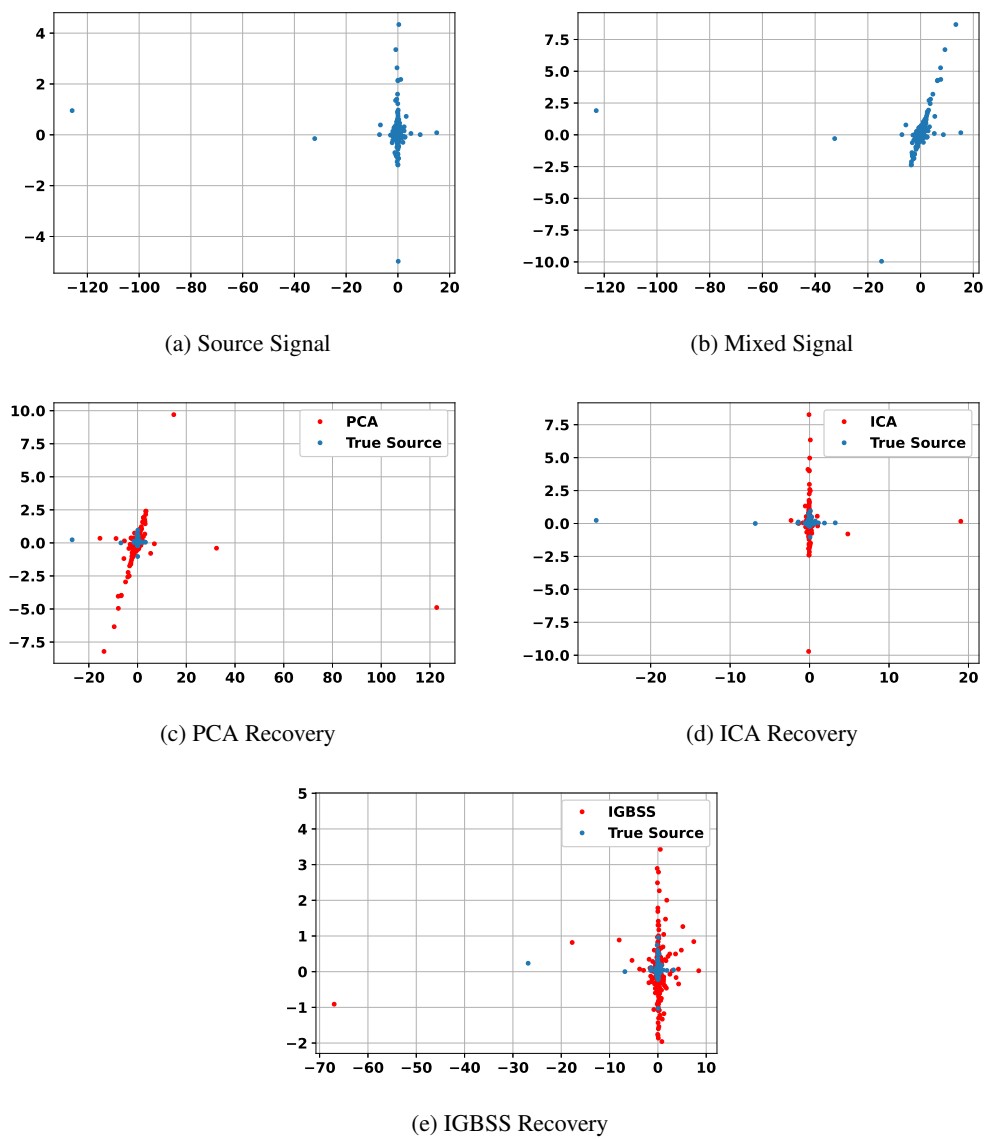

Figure 5: 2-dimensional point cloud experiment.

## A.4 Sign Inversion in ICA

When demonstrate the problem of the sign inversion in ICA. We use the same experimental set-up explained in Section 3.1 on blind source separation for affine transformation. We run the experiment on the dataset used for experimental 1 for the first order experiment and have shown the output of several runs in FastICA to show the problem of the sign inversion in Figure 7. For the 6 runs, we can see that none of the experiments were able to obtain the correct sign of the signal. This means that apply FastICA to applications where the sign of the signal is important is quite problematic.

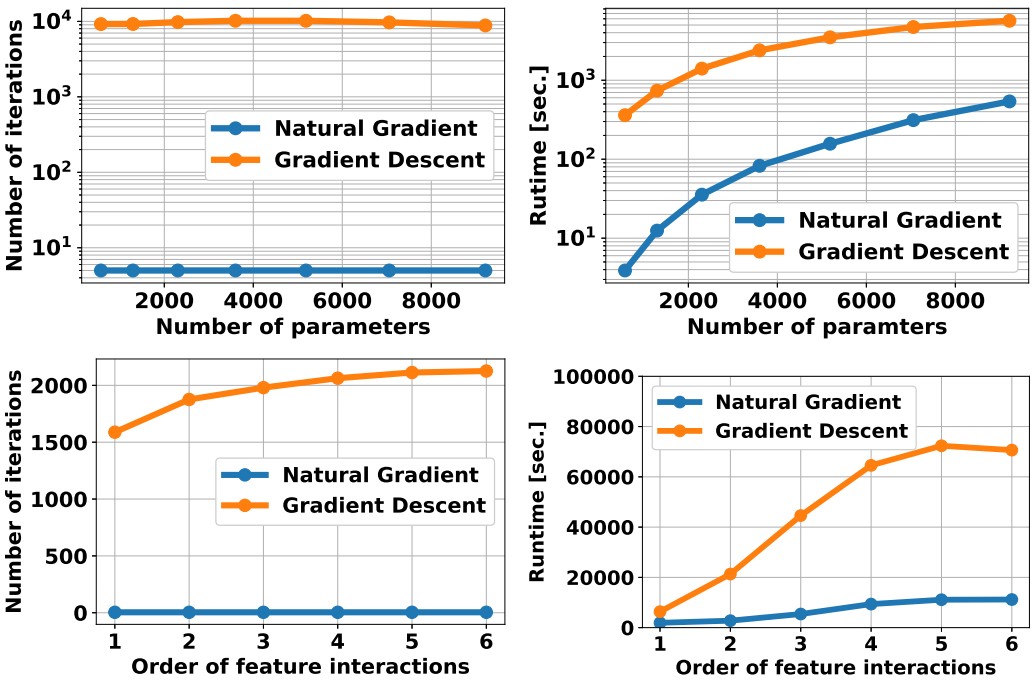

Figure 6: Experimental analysis of the scalability of number of parameters and higher-order features in the model for both natural gradient approach and gradient descent

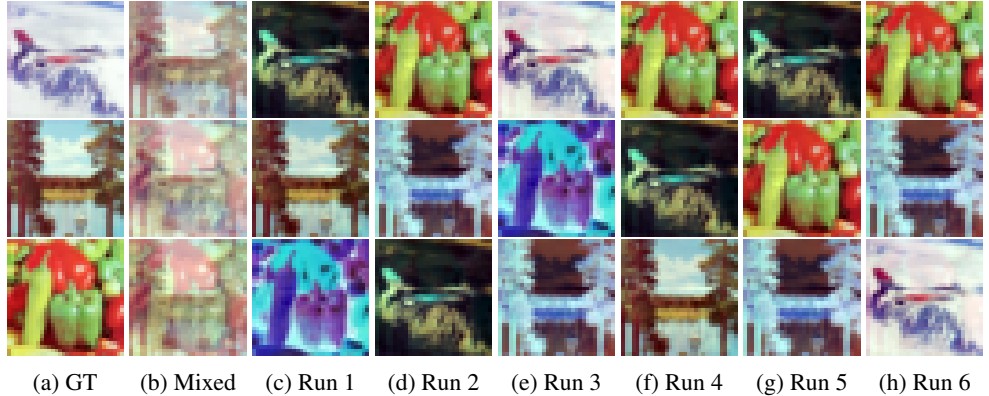

(a) GT    (b) Mixed    (c) Run 1    (d) Run 2    (e) Run 3    (f) Run 4    (g) Run 5    (h) Run 6

Figure 7: Six different runs of FastICA with the same experimental input experimental dataset as exp1 with first order interactions. The different results can demonstrate that the FastICA model is non-convex leading to potential problematic results such as the sign inversion.

