# OpenReview forum: "Hierarchical Probabilistic Model for Blind Source Separation via Legendre Transformation"
_ICLR.cc/2021/Conference — Reject_

### Official Review · AnonReviewer1 · 2020-10-27
**Hard to follow, important information missing**

**Rating:** 3
**Confidence:** 4

**Review:**

%% post-rebuttal %%
Though I appreciate the efforts of the authors to clarify their methodology and assumptions in their answer, these clarifications (which I still don't fully grasp or agree with) have not been reflected in the revised version. This work still needs a significant revision and, in my opinion, cannot be accepted in its current form.
%%%%%%%%%%%%


0) I reviewed an earlier version of this paper. The ICLR submission is similar to the earlier one, hence the similar (but updated) review. Generally, the paper is hard to follow and can only be understood by someone well-versed in both source separation and information geometry.

1) Presentation of the ideas is lacking (insufficient explanations, motivations) and I failed to understand the main contribution of the paper, i.e., how the concept of “information geometric log-linear model” can be applied to BSS. The log-linear model models a discrete probability distribution of ordered events. As such the authors introduce a specific ordering for BSS that I failed to understand. Eq. (2) plays a central role but details and motivation are lacking and it’s hard to understand why this ordering is interesting and why a log-linear model makes sense.

- Exemples would be welcome in Section 2.1. What is Omega ? what is an s ? what is S ? what is the meaning of Psi ? what is the intuition behind (1) ?

2) I would encourage the authors to clarify the settings in which their methods is applicable. It seems to be able to tackle undetermined methods and some forms of nonlinearity, though it's not clear why.
- What are the assumptions about the sources and mixtures ?
- How come your method can identify the sign of the sources ? What sort of prior information is available ?
- The preprocessing of real-valued data (basically applying exp or some form of normalisation) is disturbing.

3) The experiments consider artificial mixture of images and artificial mixtures of synthetical signals with little practical significance.
- Can you explain more clearly why ICA or NMF fail on the simple 3x3 image separation problem ?
- Do you have any idea of an application that could benefit from the nonlinear model in Section 3.2 ?

4) The introduction of the paper could be improved as it seems to reveal some misconceptions about blind source separation (BSS). For example PCA is not a method dedicated to the separation of orthogonal sources (such a problem is actually not identifiable). It is instead often used as a pre-processing in ICA (whitening). The introduction also sustain a confusion between feature extraction and source separation. These two different tasks can be handled by similar methods (ICA, NMF) but in very different settings. In source separation, the data X contains signal in rows (e.g., the cocktail party problem). In feature extraction, the focus is on the columns of X which contains data samples (such as short-time frames of a single-channel mixture).

---

> ### Author Response · Authors · 2020-11-22
> **Response to the comments of Reviewer #1 (1/2)**
>
> 0\) Thank you for your interest in reviewing our paper again. We have already revised and rewritten many parts in Section 2. The mathematical formulation of the current version should be self-contained, with the appropriate citations available if the reader requires more background in a specific area.
>
> 1\) As you correctly mention, Eq.(1) defines a discrete probability distribution of ordered events, and the relationship between events is realized as the partial order (equivalent to DAG), which can be also represented as a model matrix F. Our motivation is to use a specific partial order structure defined in Eq.(2) (shown in Figure 1) tailored to BSS.
>
> Eq.(2) specifies the partial ordering so that the model performs BSS. The graph structure which arises from this partial ordering is illustrated in Fig.1. From this figure, we can see that the graph structure is designed so that the connections between layers will perform a similar operation to the forward model of ICA $X = AZ$. For example, entry $a_{11}$ will be multiplied to $z_{11}$ and $z_{12}$, so there is an edge connecting between the mixing layer and the source layer. $z_{11}$ and $z_{21}$ is used to reconstruct $x_{11}$, so there is an edge connecting these nodes between the source layer and the received layer. These edge connections define the model matrix F in our log-linear model introduced in Section 2.1. Since the partial ordering in the log-linear model is connected in the similar fashion to the Forward ICA model, when we estimate the parameters in our proposed model, we can expect a model for BSS.
>
> 1a) These notations are all well defined just under Eq.(1). $\Omega$ is the set of possible states, using Fig.1 as an example, this is {$a_{11}, a_{12}, a_{21}, a_{22}, z_{11}, z_{12}, z_{21}, z_{22}, x_{11}, x_{12}, x_{21}, x_{22}$}.
>
> $\mathcal{S}$ is the parameter space which is a subset of $\Omega$. Using Fig.1 as an example, this is {$a_{11}$, $a_{12}$, $a_{21}$, $a_{22}$, $z_{11}$, $z_{12}$, $z_{21}$, $z_{22}$}. This is because we learn the parameters $\theta$ in $\mathcal{A}$ and $\mathcal{Z}$ but fix the parameters $\theta$ in $\mathcal{X}$ to $0$, because $\mathcal{X}$ is used to represent our input (i.e., the mixed signal), therefore $\mathcal{X} \subseteq \Omega$ but $\mathcal{X} \not \in \mathcal{S}$.
>
> $\psi$ is the partition function (normalization term), which is represented by the left most node in Fig.1. The partition function is commonly used in many machine learning and statistical models, particularly in energy based models and graphical models such as Markov Random Fields. This term is to ensure that the probability density sums to 1.
>
> 2\) We have not claimed that we are able to use our approach for undetermined methods or non-linearity directly. This is open ended as a possibility for a future research direction and is a non-trivial extension to our work, but has certainly not been mentioned or demonstrated in our paper. We have claimed and demonstrated that our model is able to separate a signal with higher-order interaction effects by including additional nodes in $\mathcal{A}$ to represent the higher-order interaction effects as explained in Section 3.2. This is different from undetermined methods or non-linearity.
>
> 2a) The assumption made between the mixed signal and source signal is that the signal can be decomposed by the relationship defined in the partial order structure shown in Eq.(2).  We also assume that the log probability density can be represented by a linear model. This implies that the probability density is within the exponential family.
>
> 2b) Classical approaches such as ICA measure how far away each signal is from a Gaussian distribution. A Gaussian distribution is symmetric so if you invert the sign, we arrive at the same point. Our approach does not have such symmetric assumptions and tries to directly recover the signal values in a similar fashion to NMF which does not have any issues in recovering the sign of the signal.
>
> The only prior information available is the mixed signal. In our case, we have assumed that our system is not underdetermined, that is $L >= N$, where the $L$ mixed signals are assumed to be linearly independent from each other.
>
> 2c) The exponential kernel to transform the data has no significant effect on the performance of the model as demonstrated in the Time Series Experiment in Section 3.3. We perform this pre-processing step because the inputs to our model require all positive values. We apply the reverse transform to our recovered signal so that it becomes the same scale as the input signal. Therefore the scale of the values of the input and output remains the same. That is, there are still negative values in the signal. This technique is widely used when using NMF for BSS.

---

> > ### Author Response · Authors · 2020-11-22
> > **Response to the comments of Reviewer #1 (2/2)**
> >
> > 3\) We have used synthetic data for our experiments because we are able to quantitatively analyse the results of our experiment using ground truth signals.
> >
> > 3a) ICA and NMF fail on the problem because they assume an affine transformation and are unable to consider the higher-order interaction effects between signals.
> >
> > 3b) We have included higher-order interaction effects into the model which typically can be found in many classical statistical models. This is different from introducing non-linearity into the model. As you can see in the equation in Section 3.2, we still have a linear assumption. There is no non-linear transformation. Multiplying signals together to consider the higher-order interaction effect is not a non-linear transformation. In any real application, we would expect higher-order interaction effects between the signals. However, the majority of the models oversimplifies this and only assumes first-order interaction.
> >
> > 4\) You are correct that the most common application for PCA is dimensionality reduction and is applied to ICA to ensure that an input matrix is not over-complete. However, the latent space in PCA can be thought to be the same as that in ICA used to represent source signals. As you mentioned, if you apply PCA on rows instead of columns, this becomes a solution for BSS. A simple search will show that there is substantial literature in applying PCA to BSS. In our revised version we have additional experiments in Appendix A.2 to demonstrate that our approach IGBSS can also be used in dimensionality reduction/feature extraction on a 2D point cloud so that the comparison between PCA, ICA and IGBSS is more clear.

---

### Official Review · AnonReviewer2 · 2020-10-27
**Paper not acceptable**

**Rating:** 2
**Confidence:** 5

**Review:**

We find he paper inappropriate for ICLR for many reasons. We give below some elements to help the authors in increasing the relevance of their work and paper.

The title is inappropriate, as Legendre transformation is not used in this work. And the word Legendre is never used in the paper.

The introduction is too long, describing many elements that are not necessary for the paper. For example, the paragraph describing the PCA and some of its variants, as well as the orthogonality constraint. In the paper, the PCA is not considered, neither the orthogonality constraint.

The authors focus on the fact that other methods are sensitive to initialization and they may converge to local optimum. In practice, the proposed method may have similar issues, as can be seen from algorithm 1 because it requires the initialization of \theta_s. It is not clear what is the used initial values, but it seems that it is 0 and not random because it makes the obtained results constant after 49 runs. This doesn’t mean that it is independent of the initialization.

It is easy to demonstrate that the proposed method cannot jointly verify the independence on initialization and the convexity/convergence to global optimum. To show this, we can initialize with the results obtained from FastICA. As FastICA sometimes outperforms all the other methods, this means that the proposed method should provide better results that FastICA throughout iterations, because the optimization problem is supposed to be convex and it converges to the global optimum; However, this means that the method depends on the initialization. If the proposed method is supposed to be invariant to initialization, this means that it will yield the same values given in the paper, thus not outperforming FastICA is several cases. This means that it converges to an optimum that is not a globale one, as FastICA converges to better optimum solution.

Experiments are not convincing, because comparing only to too old methods, such as FastICA from 2000 and dictionary learning from 1997. The authors need to compare to more recent methods from the state of the art.

There are many spelling and grammatical errors, such as disctionary.

---

> ### Author Response · Authors · 2020-11-22
> **Response to the comments of Reviewer #2**
>
> Thank you for your detailed comments. We would like to address each of your concerns in the following:
>
> 1\) Legendre transformation provides the connection between the two coordinate systems $\theta$ and $\eta$ used for optimization explained in Section 2.3. We have added an explanation to the Legendre Transformation just after Eq.(4). More details on this can be found in any introductory text to information geometry such as [Amari’16, Section 1.4].
>
> 2\) The introduction is to give background and context on what models are typically used for BSS.
>
> 3\) The statistical manifold of the exponential family studied in information geometry is always convex (please see [Amari 16, Section 1.3] for the theoretical proof) and our problem in Eq.(3) becomes always a convex optimization because of this geometric structure. Therefore, no matter what you initialize the weights to be at the beginning, after minimizing the KL divergence, we always arrive at the global optimal.
>
> 4\) Your statement is not correct. Formulating a problem as a convex optimization does not mean it always achieves the best results. Convex optimization means that our model always arrives at the best results with respect to a given objective function, which can be different across methods. Yes, you are correct in saying FastICA, sometimes performs better than our proposed approach. However, it is not correct to say that a convex model will always outperform a non-convex model, because there are different assumptions made in the model.
>
> We can use a very simple example to explain this. Take for example the problem of image classification. A linear regression model is a convex optimization, but will perform poorly. However, a deep convolutional neural network requires non-convex optimization but performs significantly better.
>
> The key advantage of formulating it as a convex optimization is the constancy of the results to always arrive at the same solution no matter the initialization of the weights, which is critical in some applications such as robotics.
>
> 5\) The experiments are compared well established unsupervised learning techniques used for BSS. To the best of our knowledge, these are the best models that are used in the same unsupervised settings. If you are able to suggest more recent techniques which are used for BSS in an unsupervised learning setting, we would be happy to include it in our experimental results.

---

### Official Review · AnonReviewer4 · 2020-11-02
**An interesting use of information geometry in blind source separation**

**Rating:** 4
**Confidence:** 3

**Review:**

Additional comments: I have read the authors' response and the other reviews. While my initial concerns have been mostly addressed, there are still concerns from the other reviews and I have revised my score accordingly.

In addition, I am not completely satisfied with the authors' response concerning higher-order interactions. In the end, even if you add other nodes, the operations are simply matrix operations and so you can model any higher order interactions with a single matrix. This should be clarified further.

______________________________________________________________________________
This paper uses an information geometry-based model to perform blind source separation. The problem is definitely significant and should be of interest to conference attendees. The approach appears to overcome many of the weaknesses in existing approaches and outperforms them in selected experiments.


Pros:
IGBSS, the proposed approach, has many advantages such as convexity and potential nonlinearity allowing it to adapt to nonlinear mixing effects. The empirical results shown here also look good.

The paper appears to be technically correct.

Cons:
The experiments seem pretty simple. Only a few experiments are performed. While the results are fairly convincing, it seems like a lot more experiments could have been performed with summary statistics reported.

At the bottom of page 3, it's stated that the partial ordering allows for higher order interactions. This isn't clear to me as it still appears that only matrix operations are being performed with this model and thus only first order interactions can be modeled. Can the authors clarify this?

I would be willing to raise my score if these issues were properly addressed.

Other comments:
In Table 1, FastICA performs best wrt RMSE in experiment 2 with third order interactions and wrt SNR in experiment 2 with second order interactions. The authors should fix this.

FastICA tends to do better wrt RMSE and SNR as the order increases. Do the authors have any insight into why this is?

---

> ### Author Response · Authors · 2020-11-22
> **Response to the comments of Reviewer #4**
>
> Thank you for your detailed comments. We would like to address each of your concerns in the following:
>
> 1\) In our revised version we have included additional experiments on a 2D point cloud in appendix A2.
>
> 2\) Our graph structure allows us to include higher-order interaction effects into our model. Higher-order interactions effects can be included in our model as explained in the last paragraph of Section 3.2, where additional nodes $a_{ln_1n_2}$ can be added to the mixing layer to include higher-order interaction effects. If we take for example, for $k=2$, we will connect node $a_{ln_1n_2}$ to the two corresponding nodes in the source layer, that is $z_{n_1}$ and $z_{n_2}$.
>
> Majority (not all) of the BSS models require a square matrix. This implies $L=N$, where $L$ is the number of mixed signals and $N$ is the number of source signals. If we introduce higher-order feature interactions the matrix becomes non-square and then several issues arise, for example in ICA, the determinant is non-trivial to compute for a non-square matrix. For these situations, it is quite typical to project the problem into lower dimensional space using PCA to ensure that the matrices are square, so that the determinant of the matrix can be computed. However, if we have used PCA to project it down into a lower dimensional space, our model no longer includes the higher-order interactions effects.
>
> >Other comments 1
>
> We do not see any error in our experimental results to fix in Table 1. We have set up different experiments, and this is the correct result of our experiment. Using SNR as a metric, FastICA performs marginally better in some cases. Please note, even if FastICA has performed better, it should be noted, our model still has its advantages in always being able to recover the same signal with different weight initialization and it is also about to recover the sign of the signal.
>
> >Other comments 2
>
> The experimental results are not comparable between different orders, because the experimental set up is different. Experiments within the same order of interactions will have used the same mixing matrix to generate the mixed signal. However, across different orders we have used different mixing matrices to general the mixed signals. Therefore experiments with different order of interactions are not comparable because we have used a different min-max normalization. However, within the same order of interactions, they are comparable.

---

### Official Review · AnonReviewer3 · 2020-11-03
**Weak discussion on the superiority of the proposed method**

**Rating:** 4
**Confidence:** 4

**Review:**

##########################################################################

Summary:


This paper formulates blind source separation as a statistical estimation problem in a log-linear model and solves the statistical problem by a modified natural gradient method.


##########################################################################

Reasons for score:


This is a complete work. However, this paper simply presents the method and experimental results and an in-depth discussion on why the proposed method should be superior than existing ones is lacking, so I do not feel this work meets the standard of ICLR. There are some misconceptions in the arguments of this paper (see cons).


##########################################################################Pros:


Pros:

1. The idea of formulating blind source separation as statistical estimation in a log-linear model seems to be novel.


##########################################################################

Cons:


1. The novelty seems limited. The main novelty is in formulating the blind source separation problem as a statistical estimation problem. The log-linear model and natural gradient method seems to be already known to experts.

2. The presentation is highly imbalanced. While the log-linear model and natural gradient method are already known, this paper spent most of the space on stating the known results and lacks an in-depth discussion on why the proposed algorithm should be superior.   The main argument supporting the proposed algorithm is that non-convex optimization is bad and convex optimization is good. This is apparently not true in general.

3. The paper argues that existing non-convex approaches are not favorable because they are sensitive to initializations. This is in general not true. The authors should show empirically that sensitivity w.r.t. initializations actually poses an issue in existing approaches, or give appropriate citations supporting their claim.

4. The paper also argues that existing approaches are not favorable because they impose strong assumptions on the problem. However, I think the log-linear model assumption in this paper is not obviously weaker than the assumptions in existing approaches.

5. This paper proposes using natural gradient descent to solve the convex optimization problem, because the iteration complexity of natural gradient descent is low. However, a low iteration complexity is not very meaningful; what’s meaningful is for example the overall computation complexity to achieve an \varepsilon-approximate solution.

6. To reduce the per-iteration complexity of natural gradient descent, this paper proposes a seemingly heuristic variant in (6) and (7) and claims the algorithm also converges to the optimum. I do not see why the claim is true.

7. The presentation can be improved. The introduction to the log-linear model and natural gradient descent is quite long but not very readable to a non-expert in the two topics. It is better to have this paper checked by a native speaker, though the number of typos and grammar mistakes is acceptable.

8. The title features “Legendre transformation,” but I do not find it in the main text.

9. Typo: The set \mathfrak{B} as defined in Section 2.3 is not exactly the set of all probability distributions.


##########################################################################

After reading the rebuttal:


I keep my score. The authors should rethink how to claim the novelty in a concrete way. The presentation needs to be improved for readers not familiar with information geometry (they should be the majority). I do not get 6) in the rebuttal.


#########################################################################

---

> ### Author Response · Authors · 2020-11-22
> **Response to the comments of Reviewer #3**
>
> Thank you for your detailed comments. We would like to address each of your concerns in the following:
>
> 1\) The log-linear model provides a relationship between a graph structure and the statistical framework. Our novelty is in designing a hierarchical graph structure for BSS problems which have not been studied in the past. In our paper, we propose the first formulation of the log-linear model which has hierarchical layers which represent the mixing layer, the source signal and the received signal. This formulation is non-trivial as the partial order structure in Eq.(2) is non-trivial.
>
> 2\) Your statement is not correct. Majority of the equations are used to formulate our BSS problem which cannot be found in literature. Majority of our equations are derived based on the partial order structure specified in Eq.(2), which leads to different equations to update the parameters $\theta$ and $\eta$.
>
> 3\) We do not claim that non-convex optimization is bad, in our experimental results we have shown that FastICA, a non-convex model, actually outperforms our proposed convex formulation for some cases. A convex formulation is highly desirable in some applications such as robotics where we are required to consistently recover the same signal when given different weight initializations.
>
> 4\) The log-linear model naturally arises from the graph structure shown in Fig.1. We are able to flexibly design the interaction between layers or change how many parameters are in the mixing matrix. We have demonstrated this capability in Section 3.2.
>
> 5\) You are correct. We already have a discussion about this in the appendix A.3. The general ballpark number that we use, if there are less than 15000 parameters, we apply natural gradient, if it is greater than 15000, it will be faster to use gradient descent.
>
> 6\) The optimum occurs when the $D(p||\hat{p}) = \eta - \hat{\eta} = 0$. This only occurs if and only if $\eta = \hat{\eta}$. This is the same when using natural gradients. Since natural gradient converges if and only if  $\eta = \hat{\eta}$, our proposal in Eq.(6) and (7) does not change the result as these update stops only if $\eta = \hat{\eta}$ is satisfied.
>
> 7\) We understand that the log-linear model and the natural gradient is not easy to understand. We have already included in Appendix A.1 the expanded version of these equations to make it easier for the reader to understand.
>
> 8\) Legendre transformation is used for the derivation of the dual coordinate system, which we have cited [Sugiyama‘17, Amari‘16]. We have included the word in the revised version just below Eq.(4).

---

### Decision · Program_Chairs · 2021-01-07
**Final Decision**

**Decision:**

Reject

**Comment:**

The focus of the submission is blind source separation (BSS). The authors propose a log-linear model based formulation to tackle the task and to relax assumption/restrictions (linear mixing, non-convex objective, ...) present in previous techniques. They use the maximum likelihood approach [Eq. (3)] with natural gradient descent for optimization, and illustrate the efficiency of the approach in two toy examples (separation of mixed images and that of sin/sign/sawtooth signals).

BSS is an important task in machine learning with various applications. As assessed by the reviewers, however, the submission is in a quite preliminary stage:
(i) Section 1 is rather long, still it lacks providing relevant context to the work.
(ii) The introduction of the main ideas/motivation, the assumptions imposed, and the explanation of the notations are missing.
(iii) The usefulness of the proposed approach is questionable; the demos focus on artificial toy examples.
More work and significant revision are needed before publication.